# Genome-Wide Association Study of Fluorescent Oxidation Products Accounting for Tobacco Smoking Status in Adults from the French EGEA Study

**DOI:** 10.3390/antiox11050802

**Published:** 2022-04-20

**Authors:** Laurent Orsi, Patricia Margaritte-Jeannin, Miora Andrianjafimasy, Orianne Dumas, Hamida Mohamdi, Emmanuelle Bouzigon, Florence Demenais, Régis Matran, Farid Zerimech, Rachel Nadif, Marie-Hélène Dizier

**Affiliations:** 1Université Paris-Saclay, UVSQ, Univ. Paris-Sud, INSERM, Équipe d’Epidémiologie Respiratoire Intégrative, CESP, 94807 Villejuif, France; mioraandria3@gmail.com (M.A.); orianne.dumas@inserm.fr (O.D.); rachel.nadif@inserm.fr (R.N.); 2Université Paris Cité, INSERM, UMR 1124, Group of Genomic Epidemiology and Multifactorial Diseases, 75006 Paris, France; patricia.jeannin@inserm.fr (P.M.-J.); hamida.mohamdi@inserm.fr (H.M.); emmanuelle.bouzigon@inserm.fr (E.B.); florence.demenais@inserm.fr (F.D.); marie-helene.dizier@inserm.fr (M.-H.D.); 3Univ. Lille, ULR 4483—IMPECS, 59000 Lille, France; regis.matran@univ-lille.fr (R.M.); farid.zerimech@chu-lille.fr (F.Z.); 4CHU Lille, 59000 Lille, France; 5Institut Pasteur de Lille, 59000 Lille, France

**Keywords:** fluorescent oxidation products, oxidative stress, genome-wide association study, chronic diseases, asthma, smoking

## Abstract

Oxidative stress (OS) is the main pathophysiological mechanism involved in several chronic diseases, including asthma. Fluorescent oxidation products (FlOPs), a global biomarker of damage due to OS, is of growing interest in epidemiological studies. We conducted a genome-wide association study (GWAS) of the FlOPs level in 1216 adults from the case-control and family-based EGEA study (mean age 43 years old, 51% women, and 23% current smokers) to identify genetic variants associated with FlOPs. The GWAS was first conducted in the whole sample and then stratified according to smoking status, the main exogenous source of reactive oxygen species. Among the top genetic variants identified by the three GWAS, those located in *BMP6* (*p* = 3 × 10^−6^), near *BMPER* (*p* = 9 × 10^−6^), in *GABRG3* (*p* = 4 × 10^−7^), and near *ATG5* (*p* = 2 × 10^−9^) are the most relevant because of both their link to biological pathways related to OS and their association with several chronic diseases for which the role of OS in their pathophysiology has been pointed out. *BMP6* and *BMPER* are of particular interest due to their involvement in the same biological pathways related to OS and their functional interaction. To conclude, this study, which is the first GWAS of FlOPs, provides new insights into the pathophysiology of chronic OS-related diseases.

## 1. Introduction

Oxidative stress (OS) was defined in 1985 as “a disturbance in the pro-oxidant/antioxidant balance in favour of the former” [1]. Beyond its essential role in life processes, OS is involved in the pathophysiology of several chronic diseases, including cardiovascular diseases, chronic kidney diseases, and asthma [2]. Sources of reactive oxygen species include the diseases themselves, through their intracellular metabolisms, and some exogenous sources, among which the most important is cigarette smoke [3].

Among the numerous biomarkers related to OS [1,4], Fluorescent Oxidation Products (FlOPs), which reflect a global measurement of oxidation of lipids, proteins, carbohydrates, and DNA [5], is of growing interest for epidemiological studies [6], as an easily quantifiable and stable biomarker of damage due to OS. The FlOPs level was found to be associated with chronic diseases such as coronary heart disease (CHD) (e.g., incidence of CHD among men without previous cardiovascular events, and risk of future CHD in healthy women) [7,8], and chronic kidney diseases [9]. Recently, we reported that high FlOPs level was associated with asthma attacks, the use of any asthma treatment, and poor control of asthma and was a predictor of asthma evolution in adults from the French Epidemiological study on the Genetics and Environment of Asthma (EGEA) [10,11]. We also found that smoking increased the FlOPs level [11].

Understanding the aetiology of multifactorial and heterogeneous chronic diseases is challenging. We hypothesised that genome-wide association studies (GWAS) of FlOPs may provide new insights into the pathophysiology of complex chronic diseases related to the OS pathway, including asthma. Indeed, previous GWAS of levels of circulating protein biomarkers related to chronic obstructive pulmonary disease, another chronic lung disease, was useful to identify new genes linked to this disease [12]. Such an approach was also useful to identify new genes through the study of biomarkers of other chronic diseases such as kidney or cardiovascular diseases [13,14]. To our knowledge, no GWAS of biomarkers related to the OS pathway, and in particular FIOPs, has been published to date.

Taking advantage of the extensive biological, genotypic, and phenotypic characterisation of >1000 adults from the EGEA study, we aimed to identify genetic loci associated with the FlOPs level through a GWAS. As smoking is the main environmental source of OS and is associated with the FlOPs level, we performed two supplementary GWAS analyses separately in contrasted groups according to tobacco smoking status that could help us to identify genetic loci that could have been missed in the whole sample.

## 2. Materials and Methods

### 2.1. Study Population (EGEA Study)

EGEA is a French cohort study based on an initial group of asthma cases recruited in five chest clinics from Grenoble, Lyon, Marseille, Montpellier, and Paris, along with their first-degree relatives, and population-based controls (EGEA1, 1991–1995). The protocol of the study has been described previously [15,16]. Briefly, the asthma cases and their first-degree relatives were recruited from respiratory or allergic clinics. The adult cases were recruited in the five cities, and the child cases were recruited in Paris, Grenoble, and Marseille. Control adults were recruited from electoral rolls in Paris, Lyon, Montpellier, and Grenoble, a check-up centre in Marseille, and surgery clinic from the same hospital in Paris and Grenoble. Control children were always recruited from surgery clinics. Overall matching by month of exam, age decade, sex, and centre was done. A first follow-up of the initial cohort was conducted between 2003 and 2007 (EGEA2) including 1602 participants (98% adults) with complete examination. At each survey, all participants answered standardised and validated questionnaires to identify asthma and to determine respiratory and allergic symptoms, treatments, environmental exposures, and lifestyle characteristics, including tobacco smoking status. More details are given in Appendix A. The data used for the present analyses were elicited at EGEA2.

The EGEA collection was certified ISO 9001 from 2006 to 2018 [17]. All participants signed a written informed consent, and ethical approval was obtained from the relevant institutional review board committees (Cochin Port-Royal Hospital and Necker-Enfants Malades Hospital, Paris, France).

### 2.2. FlOPs Level

Plasma samples were collected in EGEA2 between 2003 and 2006 and stored immediately at −80 °C during 5.0 to 8.0 years until FlOPs measurements. The plasma FlOPs level was measured as previously described [7]. Briefly, plasma was extracted into a mixture of ethanol/ether (3/1 *v*/*v*) and measured using a spectrofluorometer (360 nm excitation wavelength, 430 nm emission wavelength). Fluorescence was expressed as a unit of relative fluorescence intensity (RFU)·mL^−1^ of plasma. Each sample was replicated. The intra-assay coefficient of variation (CV) for FlOPs was less than 20%. The dosages for which the CV were ≥20 % or those that were haemolysed were removed of analysis (n = 11 and n = 8, respectively, see Appendix A).

### 2.3. Genotyping

The EGEA participants were genotyped using Illumina 610 Quad array at the Centre National de Génotypage (CNG, Evry, France) as part of the European Gabriel consortium asthma GWAS [18]. As part of this consortium, principal-components analysis was conducted for all participants to control population admixture and was carried out using the EIGENSTRAT2.0 software. Putative non-European samples were flagged as outliers and eliminated from subsequent analyses [18]. Stringent quality control (QC) criteria were used to select both individuals and genotyped Single Nucleotide Polymorphisms (SNPs) for analysis [19]. Among participants with genotyped data (n = 1481), 44 with invalid genotyped data were excluded (see Appendix A, flow chart, for details). The following SNPs quality controls were applied: genotyping call rates ≥ 97% and departure from the Hardy–Weinberg equilibrium in the controls (*p*-value ≥ 1.0 × 10^−4^) and minor allele frequencies (MAF) ≥ 5%. After this SNPs QC control, 66,422 SNPs were excluded, and a total of 501 167 SNPs were available for the analysis. To investigate regions of interest, including the top three SNPs, imputed SNPs from the reference panel 1000 Genome Phase I were used [20]. The software IMPUTE2 was used for imputation [21]. Imputed SNPs were kept for analysis if their imputation information score was greater than or equal to 0.70 and if their minor allele frequency (MAF) was greater than or equal to 0.05.

### 2.4. Definitions of Population’s Characteristics

#### 2.4.1. Asthma

Ever-asthma status was generated as a dichotomous variable (never-asthmatic/ever-asthmatic). Ever-asthmatics were participants who answered positively to at least one of the two following questions: “*Have you ever had attacks of breathlessness at rest with wheezing?*” or “*Have you ever had asthma attacks?*”, or those who were recruited as asthmatic cases at EGEA1. Never-asthmatics were those who answered negatively to the two questions above; they were not recruited as asthmatic cases at EGEA1.

#### 2.4.2. Chronic Bronchitis

Chronic bronchitis was generated as a dichotomous variable (yes/no). Participants with chronic bronchitis were those who answered positively to at least one of the two following questions: “*Do you usually cough during the day or at night in winter almost every day for three months of continued every year?*” or “*Do you usually spit during the day or at night in winter, almost every day for three months of continued every year?*”.

#### 2.4.3. Lung Function

A lung function test with spirometry and methacholine challenge was performed using standardised protocol with similar equipment across centres and according to the American Thoracic Society/European Respiratory Society guidelines [22]. The forced expiratory volume in one second (FEV_1_) percent predicted value was based on Quanjer et al. reference equations [23]. For participants with a FEV_1_ ≥ 80% of the predicted value, a methacholine bronchial challenge test was performed (maximum dose 4 mg) using a Biomedin spirometer (Biomedin Srl, Padua, Italy) in all centres, except in Lyon, where a Pneumotach Jaeger spirometer (Jaeger) was used. The following measures of lung function were used as continuous variables and expressed as %: FEV_1_ and Forced Vital Capacity (FVC). FEV_1_ was also generated as a dichotomous variable (FEV_1_ < 80%, FEV_1_ ≥ 80%).

#### 2.4.4. Smoking Status

Tobacco consumption was defined by the answer to the question “*Do you smoke or have you smoked previously one cigarette per day or more, for at least a year?*”. If so, participants were asked their age at the start of smoking and the age of quitting, if applicable. Participants were also asked to quantify the average daily consumption of cigarettes and the average weekly consumption of cigars, if applicable. Current smoking status was generated as a 3-class categorical variable: never-smoker, ex-smoker, or current smoker. Lifelong cumulative quantity of tobacco was generated as a continuous variable and also categorised using a 4-class variable, with cut-offs defined a priori: never-smokers; <10 pack-year; 10–20 pack-year; and >20 pack-year.

#### 2.4.5. Body Mass Index (BMI)

BMI was generated as a continuous variable and was also expressed as a dichotomous variable (<30 kg/m^2^; ≥30 kg/m^2^).

#### 2.4.6. Biological Parameters

Total serum Immunoglobulin E (IgE) determination was assessed by the UniCAP system (Pharmacia^®^) from blood samples in a centralised laboratory and expressed in international units (IU) per millilitre. For the analysis, total IgE level was examined as a continuous variable.

Blood neutrophil and eosinophil counts were expressed in cells/mm^3^ and coded as continuous variables [24,25].

### 2.5. Statistical Methods and Strategy of Analysis

#### 2.5.1. Characteristics of the Studied Population and Association with the FlOPs Level

First, characteristics of the studied population were described and summarised as n (%) or mean (m) ± standard deviation (sd), according to the type of variable, either quantitative or qualitative. Due to their skewed distribution, the FlOPs level was log-transformed and expressed as geometric mean (GM) and values of first quartile (Q1) and third quartile (Q3).

In order to select potential confounding factors prior to GWAS, we estimated associations between log-FlOPs and several characteristics of the whole sample using Gaussian linear models, taking into account EGEA family structure, by Generalised Estimated Equation (GEE, SAS v9.4 (SAS Institute, Cary, NC, USA), proc genmod, option repeated). We previously identified the following characteristics as factors associated with FlOPs: age, sex, current smoking status, lifelong cumulative quantity of tobacco, blood neutrophil count, and FEV_1_. Age was entered either as a continuous variable or a categorical one, with cut-points defined a priori (<25 years; 25–34 years; 35–44 years; 45–54 years; and ≥55 years). Regarding tobacco smoking, models included either smoking status as a categorical variable (never smokers; ex-smokers; and current smokers), or lifelong cumulative quantity of tobacco as a continuous variable or a 4-class categorical variable. The best model was selected based on the QIC, an Akaike’s Information Criterion in the framework of GEE models [26]. Among all models tested, the best model included age (continuous), sex, and lifelong cumulative quantity of tobacco (never-smokers; <10 pack-year; 10–20 pack-year; and >20 pack-year).

Adjusted log-FlOPs were obtained as residuals of the best linear model identified in the previous step. Z-scores were then obtained by standardizing residuals, and adequacy to Gaussian distribution was assessed using the Kolmogorov test. In order to exclude participants whose log-FlOPs were poorly predicted by the linear model, participants with the highest Z-score (i.e., |Z-score| > 3, corresponding to the 0.1th and 99.9th percentiles of a standard Gaussian distribution) were excluded. The process was repeated until no significant deviation from Gaussian distribution was evidenced. This process excluded 20 participants (see Appendix A).

Adjusted log-FlOPs used for GWAS stratified on smoking status (see below) were generated by applying the same procedure as described above, except that lifelong cumulative quantity of tobacco was not entered in the model.

#### 2.5.2. GWAS of the FlOPs Level

We first conducted a GWAS of the FlOPs level in the whole sample with genotyped data. Then, we conducted two supplementary GWAS separately in contrasted groups according to smoking status at the time of measurement: never-smokers and current smokers. Ex-smokers were excluded from this analysis.

An association analysis between adjusted log-FlOPs (standardised residual) and each SNP was performed by the Gaussian linear model, adjusted for principal components (PCs) to account for within European diversity. The EGEA family structure was taken into account using a robust variance estimator for clustered data (STATA command: regress, option vce(cluster), within family). SNPs were coded under an additive genetic model.

For the top three SNPs obtained by each of the three GWAS (in the whole sample, in never-smokers and in current smokers), further analyses were conducted. First, we split the sample into two actual independent sub-samples regarding the ascertainment mode (controls vs. cases/relatives) to check the consistency of the results by using a homogeneity test between the sub-samples. Second, due to the mode of ascertainment of EGEA families, i.e., through asthmatic participants, the independence of the results regarding the asthma status was verified by homogeneity test according to ever-asthma status (never-asthmatics vs. ever-asthmatics). A test for homogeneity was performed by fitting the interaction term between the SNP and dummy variables (cases/related vs. controls, and ever-asthmatics vs. never-asthmatics, respectively) in models. The top three SNPs of the three GWAS (in the whole sample, the never-smokers and the current smokers) was also focused on: we used imputed data from the reference panel 1000 Genome project Phase 1 [20] CEU population, spanning 500 kb on each side of each top SNP. For each region of interest, association results were graphically represented using LocusZoom [27].

All analyses were performed using SAS v9.4 (SAS Institute, Cary, NC, USA) or STATA v14.1 (StataCorp. 2015. Stata Statistical Software: Release 14. College Station, TX, USA: StataCorp LP). All tests were two-sided. To account for multiple testing, the Bonferroni-corrected significance *p*-value threshold applied to the Meff (effective number of independent tests after discarding dependence due to linkage disequilibrium (LD) between the SNPs) was calculated. For a chip of 610K SNPs, the significance *p*-value threshold was estimated to be 1.3 × 10^−7^ [28].

#### 2.5.3. eQTLs, meQTLs, and Functional Annotations

We investigated whether the top three SNPs (or their proxies, r^2^ ≥ 0.8) in the whole sample, in never-smokers, and in current smokers were cis-expression Quantitative Trait Loci (cis-eQTLs) or methylation Quantitative Trait Loci (meQTLs). For eQTLs and meQTLs, we used the browser Phenoscanner v2 (http://www.phenoscanner.medschl.cam.ac.uk/, accessed on 18 October 2021), which combines several databases, e.g., the Genotype-Tissue Expression project (GTEx, https://www.gtexportal.org/home/, accessed on 18 October 2021) for eQTLs, and BioSQTL and Gaunt’s databases for meQTLs) and that includes e-QTL data from many tissues [29,30,31,32]. Furthermore, functional annotations of these SNPs (or proxies) were done using the HaploReg v4.1 tool (https://pubs.broadinstitute.org/mammals/haploreg/haploreg.php, accessed on 18 October 2021). HaploReg annotates SNPs in terms of colocalisation with regulatory elements, such as promoter and enhancer marks, DNase I hypersensitivity sites, and transcription factor (TF) and protein-binding sites, based on Roadmap Epigenomics data and Encyclopedia of DNA Elements data [33]. We checked the potential deleteriousness of SNPs using the combined annotation-dependent depletion tool (CADD v1.4, https://cadd.gs.washington.edu/, accessed on 1 April 2022). The CADD tool scores the predicted deleteriousness of single nucleotide variants and insertion/deletions variants in the human genome by integrating multiple annotations [34]. Note that a CADD score ≥ 15 indicates a deleterious effect of an SNP.

## 3. Results

The present analysis was carried out among adult participants (≥16 years old) at EGEA2 with available data on the FlOPs level, valid genotyped data, and asthma and tobacco smoking status. A total of 1216 adult participants were included in the analyses (see Appendix A).

Compared to the 355 adult participants not included in the analyses, the 1216 participants did not differ in terms of age, sex, and asthma status (data not shown).

### 3.1. Characteristics of the Studied Population

The characteristics of the 1216 participants are presented in Table 1. The results are presented for the whole sample and according to asthma status (ever-asthmatics and never-asthmatics), to the current tobacco smoking status (never-smokers and current smokers), and to the study design (controls and cases/relatives). In the whole sample (mean age 43.3 years, 51% women), 44% had ever-asthma and 23% were current smokers. The geometric mean (GM) (Q1, Q3) of the FlOPs level was 92.3 (80, 105) RFU/mL.

Associations between the FlOPs level and the characteristics of the whole sample are presented in Appendix A (see Appendix A). the FlOPs level was independently associated with age, sex, and smoking (all *p*-values < 5.0 × 10^−3^): it increased significantly with age, was significantly higher in women than in men, in ex-/current smokers (GM = 97.2 and 93.8 RFU/mL, respectively) as compared to never-smokers (GM = 89.1 RFU/mL), and increased significantly with lifelong cumulated quantity of cigarettes smoked. The geometric mean (GM) (Q1, Q3) of the FlOPs level was 93.4 (81, 107), 98.7 (88, 108), and 100.3 (88, 114) RFU/mL in participants with lifelong quantity of cigarettes smoked of <10, 10–20, and >20 pack-year, respectively (*p* < 1.0 × 10^−4^).

### 3.2. GWAS of the FlOPs Level

#### 3.2.1. Whole Sample

Table 2 reports the results of the associations in the whole sample for the 10 SNPs showing the strongest signals. The Manhattan plot is available in Appendix A, and the Q-Q plot in Appendix A shows that there was no inflation in the statistical test, with the genomic inflation factor estimated to 1.002. The top three SNPs were rs270404, located in the *BMP6* gene on chromosome 6p24.3 (*p* = 3.0 × 10^−6^); rs13223298, located upstream (from 2 kb apart) of the *BMPER* gene on chromosome 7p14.3 (*p*-value = 8.7 × 10^−6^); and rs491274, located in the intergenic region nearest *SEMA6D* (from 607 kb apart) genes on chromosome 15q21.1 (*p* = 8.9 × 10^−6^). The association analysis for these top three SNPs in the two independent sub-samples (controls vs. cases/relatives) showed no indication of heterogeneity (all *p*-values > 0.6, see Appendix A).

The associations stratified by asthma status (never-asthmatics, ever-asthmatics) for the top three SNPs associated with the FlOPs level in the whole sample are reported in Table 3. No indication of heterogeneity was observed (all *p*-values ≥ 0.25).

An analysis using imputed SNPs spanning 500 kb on each side of each top SNP in the regions of the top three SNPs located in *BMP6*, near *BMPER* and near *SEMA6D,* confirmed the initial results (see Appendix A). Analyses of imputed SNPs in these regions found signals with similar or slightly improved significance levels at genotyped SNPs, and for at least two other imputed SNPs, which were close to and in strong linkage disequilibrium (LD, r² > 0.8) with the genotyped top SNP (Appendix A).

#### 3.2.2. In Never-Smokers and in Current Smokers

Table 4 presents the associations with FlOPs for the top 10 SNPs in never-smokers and in current smokers. Manhattan plots are available for the two GWAS in Appendix A; and the Q-Q plots in Appendix A and show that there was no inflation in the statistical test for the two GWAS, with genomic inflation factors estimated to 1.006 and 1.03, respectively.

In never-smokers, the top three SNPs (all *p*-values < 5.0 × 10^−7^) were rs17823624 located in the *COL21A1* gene on the chromosome 6p12.1 (*p* = 2.3 × 10^−7^), rs6606856 located in *GABRG3* gene on chromosome 15q12 (*p* = 4.1 × 10^−7^), and rs2962642 located in an intergenic region near *NUDT12* (from 568 kb apart) on chromosome 5q21.2 (*p* = 4.4 × 10^−7^). For these top three SNPs, association analysis in the two independent sub-samples (controls vs. cases/relatives) showed no indication of heterogeneity (all *p*-values > 0.6, see Appendix A). Besides that, association analysis for these top SNPs yielded similar results in never-asthmatics and in ever-asthmatics, with no indication of heterogeneity (*p* ≥ 0.10, Table 5). Note that none of the top three SNPs found in never-smokers showed indication of association in current smokers (all *p*-values > 0.10).

In current smokers, the top three SNPs were rs3851212 located in *CRYBG1* on chromosome 6q21 (*p* = 2.4 × 10^−9^, exceeding the significance level of 1.3 × 10^−7^), rs1793958 located in *COL2A1* on chromosome 12q13.11 (*p* = 4.7 × 10^−7^), and rs17174795 located upstream *PTPRO* (from 1.4 kb apart) on chromosome 12 p12.3 (*p* = 9.2 × 10^−7^). For these top three SNPs, association analysis in the two independent sub-samples (controls vs. cases/relatives) showed no indication of heterogeneity (all *p*-values ≥ 0.4, see Appendix A). No heterogeneity of association signals was detected according to asthma status (all *p*-values ≥ 0.7, except for rs17174795 with *p*-value = 0.05, but not significant after correction for multiple testing; see Table 5). The top three SNPs found in current smokers showed no indication of association in never-smokers (*p* > 0.20) or only a weak signal (*p* = 0.05).

Analysis using imputed SNPs spanning 500 kb on each side of each top SNP in the regions of the top three SNPs in both never-smokers and current smokers confirmed the initial results, with a similar significance level as those observed with genotyped SNPs (Appendix A). Analyses of imputed data in the region around two of the top six SNPs found additional signals at imputed SNPs close to and in strong linkage disequilibrium (LD, r² > 0.8), with the genotyped top SNPs, 12 for rs2962642 located near *NUDT12* with similar significance level (Appendix A) and two for rs17174795 located near *PTPRO* with improved significance level (Appendix A). These results also supported the initial findings for these top two SNPs.

### 3.3. eQTLs, meQTLs, and Functional Annotations

Using the browser Phenoscanner v2 (http://www.phenoscanner.medschl.cam.ac.uk/, accessed on 18 October 2021), we found that top two SNPs, one in never-smokers (rs17823624 in *COL21A1*) and one in current smokers (rs1793958 in *COL2A1*), were associated with gene expression in a whole blood sample from subject of European ancestry (Appendix A). The SNP rs1783624 was associated with gene expression of *DST* (*p*-value = 2.0 × 10^−15^), while the SNP rs1793958 was associated with the expression of five genes belonging to the 12q13.11 region: *OR10AD1*, *PFKM*, *SENP1*, *TMEM106C,* and *VDR* (all *p*-values < 2.5 × 10^−8^). No eQTL was found for the other top SNPs. Furthermore, we found from 1 to 26 CpG sites considering all top three SNPs of GWAS in the whole sample, in never-smokers and in current smokers (See Appendix A). Most CpG sites were located near or in the same gene as the associated SNP. The SNP rs13223298 in *BMPER* was found to be more associated with the methylation level of one CpG site located on another chromosome than *BMPER*, in the *PTPN22* gene on chromosome 1p13.2 (*p*-value = 9.3 × 10^−8^, not significant after correction for multiple testing).

Note that a proxy of rs3851212 (the top SNP in current smokers located in *CRYBG1*) rs79231630 had a CADD score equal to 14.5. Detailed results for CADD scores are presented in Appendix A.

Using the functional annotation tool HaploReg-v4.1, we found that all top three SNPs (or their proxies) evidenced respectively in the whole sample, in never-smokers and in current smokers, mapped to marks of active regulatory elements, including cells from heart, lung, kidney, breast, and brain. Detailed results for functional annotations are presented in Appendix A (see Appendix A).

## 4. Discussion

This first genetic study on the FlOPs level identified several variants, among which those located in *BMP6*, near *BMPER* and between *SQOR* and *SEMA6D* were the most strongly associated with this biomarker in the whole sample. Stratified analyses on tobacco smoking status identified other genetic variants: among them, the top three SNPS in never-smokers located in *COL21A1*, in *GABRG3,* and near *NUDT12*, and the top three SNPS in current smokers located in *CRYBG1*, in *COL2A1,* and near *PTPRO*.

Our study is based on the hypothesis that GWAS of FlOPs may provide new insights nito the pathophysiology of chronic diseases related to the OS pathway. The GWAS analyses we performed were based on the EGEA study, whose participants had extensive clinical, genetic, biological, and environmental characterisation. To our knowledge, there was no other epidemiological study with such data for a replication sample. Interestingly, all our association results were supported by consistent results observed in controls and cases/relative sub-samples. Furthermore, analyses of imputed data in the region around each top SNP confirmed our initial association results obtained with genotyped data. However, all our findings should be validated/replicated in other independent cohorts.

Due to the ascertainment mode of families in the EGEA study, i.e., through asthmatic participants, and the involvement of the OS pathway in the pathophysiology of asthma, we repeated our analyses in ever- and never-asthmatics in order to evaluate the associations independently of the disease. The results were consistent between never- and ever-asthmatics, which showed the independence of our results from the disease. Furthermore, we verified that any of our top SNPs were associated with lung function or adult-onset asthma in EGEA sample [19]. All these results suggest that our main results are not driven by asthma.

In the whole population, the strongest association signals were observed for rs270404 located in *BMP6* and rs13223298 located near *BMPER* (i.e., 2.2 kb from that gene). *BMP6* and *BMPER* were reported to interact physically in a functional study [35]. In line with this result, we tested the effect of the statistical interaction between these SNPs on FlOPs and found a borderline significant interaction (*p*-value = 0.07). *BMP6* (Bone Morphogenetic Protein 6) encodes a secreted ligand of the transforming growth factor-beta (TGF-beta) superfamily of proteins, and *BMPER* (BMP Binding Endothelial Regulator) encodes a secreted protein that interacts with and inhibits the bone morphogenetic protein (BMP) function. It is noteworthy that these two genes belong to the biological process “regulation of pathway restricted SMAD protein phosphorylation” pathway (GO:006093) that is involved in the TGF-beta receptor signalling pathways [36]. The role of TGF-beta has been discussed in chronic asthma, as a potent fibrogenic growth factor overexpressed in the asthmatic lung [37]. Moreover, *BMPER* belongs to the biological process “immune response” pathway (GO:0006954) [38]. From the GWAS Catalog [39], we found that *BMP6* was associated with FVC [40,41,42]; FEV_1_ [42]; and, to a lesser extent, chronic obstructive pulmonary disease [43] and small cell lung carcinoma [44]. In previous GWAS, associations were reported between *BMPER* and FVC [40], FEV_1_ [45], and other chronic diseases such as Alzheimer’s disease [46] and metastatic colorectal cancer [47]. Moreover, the top two SNPs in *BMP6* and near the *BMPER* map to marks of active regulatory elements in heart, lung, brain, breast, and kidney tissues and the top SNP near *BMPER* was associated with the methylation level of one CpG site located in *PTPN22* (Protein Tyrosine Phosphatase Non-Receptor Type 22), a gene involved in “NF-Kappa B signalling” and “immune response” pathways.

The third strongest association signal was observed for rs491274 located nearest *SEMA6D* (i.e., 607 kb appart). *SEMA6D* (Semaphorin 6D) was found to be associated with parental longevity [48,49] and lung carcinoma [44] in previous GWAS.

As cigarette smoke is the most important exogenous source of ROS, we performed GWAS separately in two contrasted sub-groups according to smoking status and identified specific association signals in each sub-group. In never-smokers, the top SNP was rs1782324 located in *COL21A1* (Collagen Type XXI Alpha 1 Chain). In previous GWAS, *COL21A1* showed associations with lung function [40] and to a lesser extend with allergic sensitisation [50] and small cell lung carcinoma [44]. We also found that rs17823624 was eQTL of *DST* (Dystonine), a gene close to *COL21A1*, for which variants were found to be associated with lung function [40,42] and Alzheimer’s disease [51]. The next top SNP rs6606856 was located in *GABRG3* (Gamma-Aminobutyric Acid Type A Receptor Subunit Gamma3), a gene involved in the “response to drug” biological pathway (GO:0042493). This gene was shown to be associated with gene methylation in the lung tissue of smokers, as reported in a previous GWAS [52]. An association between *GABRG3* and several chronic diseases, including Alzheimer’s disease [53], ovarian carcinoma [54,55], and non-melanoma carcinoma [56], was also reported in other previous GWAS. Finally, the third highest SNP rs2962642 was located near (i.e., 568 kbp apart) to *NUDT12* (Nudix Hydrolase 12). This gene was involved in the “NADH metabolic process” biological pathway (GO:0006734), and was shown to be associated with longevity [57] and smoking behaviour [58] in previous GWAS. Furthermore, proxies of rs2962642 map onto the regulatory motifs altered for histone deacetylase 2 (HDAC2), whose activity is regulated by oxidative stress.

In current smokers, the top SNP, rs3851212, was located in *CRYBG1*, which exceeded the genome-wide significance threshold level of 4.3 × 10^−8^ when accounting for the three GWAS. The role of *CRYBG1* (crystallin beta-gamma domain containing 1), also called *AIM1* (absent in melanoma 1 protein) in malignant melanoma, is well-known [59]. To note, rs3851212 is located 50 kb from *ATG5* (autophagy-related 5), which belongs to the biological pathway “immune system process” (GO:0002376) and is involved in mitochondrial quality control after oxidative damage, and in subsequent cellular longevity. In previous GWAS, *ATG5* was found to be associated with allergic diseases [60], including asthma [61,62,63,64], and with several other chronic diseases such as systemic lupus erythematous [65,66,67,68], rheumatoid arthritis [69,70], and multiple myeloma [71]. Furthermore, note that a proxy of rs3851212—rs79231630—has a CADD score close to 15, indicating deleterious effect of the SNP. The next top SNP is rs1793958, located in *COL2A1* (collagen type II alpha 1 chain), a gene involved in “regulation of immune response” biological process (GO:0050776). *COL2A1* was found to be associated with rheumatoid arthritis [72] and with prostate carcinoma [73,74,75] in previous GWAS. On the other hand, the third highest SNP rs17174795 is located 1.4 kb apart from *PTPRO* (Protein Tyrosine Phosphatase Receptor type O), which has been suggested as a candidate tumour suppressor via the NF-Kappa B signalling pathway [76,77], and the transcription factor NF-Kappa B plays a central role in inflammatory airway diseases such as asthma [78].

None of the top signals found in one sub-group of smoking status were found in the other sub-group, nor in the whole sample, showing that, as we hypothesised, accounting for smoking status may help one to identify loci not found in the whole sample. These results are likely explained by the existing interactions between the environment (here smoking) and genes that lead to “up” or “down” regulation of the pathways that influence the level of FlOPs.

None of the top signals found in the whole sample were found in the two contrasted sub-groups according to smoking status, which is an interesting result. Indeed, these analyses were carried out with the objective to identify genetic loci that could have been missed in the whole sample as smoking is the main environmental source of OS and is associated with FlOPs. The two GWAS in groups contrasted by smoking revealed additional genetic loci to those found in the whole sample.

Overall, many of the top SNPs identified by the three GWAS are located in regions comprising promising candidate genes. Among them, *BMP6*, *BMPER*, *GABRG3*, and *ATG5* are the most relevant because of both their link to biological pathways related to OS and their association with several chronic diseases, for which the role of OS in their pathophysiology has been pointed out. *BMP6* and *BMPER* are of particular interest due to their involvement in the same biological pathways related to OS and their functional interaction.

## 5. Conclusions

In conclusion, the present study identified, for the first time, new and promising candidate genes associated with the FlOPs level potentially involved in the pathophysiology of chronic diseases through their link with the oxidative stress pathway. Although further studies are needed to replicate these findings, this work highlights the interest in performing genome-wide analyses of biomarkers to identify new genes and potential mechanisms related to specific pathways common to chronic diseases.

## Figures and Tables

**Table 1 antioxidants-11-00802-t001:** The characteristics of the studied population.

	Whole Sample	Never- Asthmatics	Ever- Asthmatics	Never- Smokers	Current Smokers	Controls	Cases/Relatives
	(n = 1216)	(n = 684)	(n = 532)	(n = 604)	(n = 275)	(n = 243)	(n = 973)
Age, m ± sd	43.3 ± 16.4	46.3 ± 15.9	39.4 ± 16.4	42.7 ± 16.9	35.5 ± 14.1	47.2 ± 17.2	42.3 ± 16.1
Sex, women, n (%)	621 (51.1)	374 (54.7)	247 (46.4)	338 (56.0)	131 (47.6)	125 (51.4)	496 (51.0)
Current smoking status, n (%)							
Never-smokers	604 (49.7)	342 (50.0)	262 (49.2)	604 (100)	-	114 (46.9)	490 (50.4)
Ex-smokers	337 (27.7)	202 (29.5)	135 (25.4)	-	-	70 (28.8)	267 (27.4)
Current smokers	275 (22.6)	140 (20.5)	135 (25.4)	-	275 (100)	59 (24.3)	216 (22.2)
Smoking, pack-year, n (%)							
Never-smokers	604 (49.7)	342 (50.0)	262 (49.3)	604 (100)	-	114 (46.9)	490 (50.4)
<10	384 (31.6)	197 (28.8)	187 (35.1)	-	186 (67.6)	75 (30.9)	309 (31.8)
10–20	111 (9.1)	68 (9.9)	43 (8.1)	-	44 (16.0)	23 (9.5)	88 (9.0)
>20	117 (9.6)	77 (11.3)	40 (7.5)	-	45 (16.4)	31 (12.7)	86 (8.8)
BMI, kg·m^−2^, n (%)	n = 1201	n = 676	n = 525	n = 597	n = 272	n = 240	n = 961
≥30	118 (9.8)	66 (9.8)	52 (9.9)	56 (9.4)	16 (5.9)	20 (8.3)	98 (10.2)
IgE	n = 1214	n = 682	n = 532	n = 603	n = 274	n = 242	n = 972
IgE, IU/mL, m ± sd	222 ± 453	134 ± 352	333 ± 537	198 ± 431	304 ± 528	116 ± 258	248 ± 487
White blood count	n = 1206	n = 677	n = 529	n = 601	n = 273	n = 242	n = 964
Neutrophils, cells/mm^3^, m ± sd	3986 ± 1391	3959 ± 1317	4020 ± 1481	3879 ± 1278	4314 ± 1535	3985 ± 1317	3986 ± 1410
Eosinophils, cells/mm^3^, m ± sd	202 ± 155	168 ± 125	244 ± 179	203 ± 166	220 ± 162	160 ± 116	212 ± 162
Lung Function	n = 1197	n = 673	n = 524	n = 594	n = 272	n = 238	n = 959
FEV_1_, % predicted, m ± sd	103 ± 17.8	107 ± 16.3	97 ± 18.2	104 ± 17.5	100 ± 15.6	105 ± 15.4	102 ± 18.3
FVC, % predicted, m ± sd	110 ± 17.1	112 ± 17.5	108 ± 16.3	112 ± 17.1	107 ± 15.0	109 ± 15.3	111 ± 17.5
FEV_1_ ≥ 80%, n (%)	1097 (91.7)	644 (95.7)	453 (86.5)	556 (93.6)	253 (93.0)	228 (95.8)	869 (90.6)
Chronic bronchitis, n (%)	n = 1205	n = 679	n = 526	n = 597	n = 274	n = 241	n = 964
Yes	114 (9.5)	39 (5.7)	75 (14.3)	47 (7.9)	40 (14.6)	19 (7.9)	95 (9.9)
FlOPs, RFU/mL, GM (Q1, Q3)	92.3 (80, 105)	93.9 (82, 108)	90.4 (78, 102)	89.1 (77, 101)	93.8 (82, 107)	92.6 (81, 107)	92.3 (79, 105)

m, mean; sd, standard deviation; FEV_1_, forced expiratory volume in one second; FVC, force vital capacity; GM, geometric mean; Q1, first quartile; and Q3, third quartile.

**Table 2 antioxidants-11-00802-t002:** The top 10 SNPs associated with the FlOPs level in 1216 adults from the EGEA study.

Chr	Gene	Nearest Gene	Genomic Location	Marker	Position bp (hg38)	Band	A1/A2	EAF	Beta ± se	*p*
6	*BMP6*	*TXNDC5*	Intronic	rs270404	7,757,141	p24.3	A/G	0.41	−0.20 ± 0.04	3.0 × 10^−6^
7		*BMPER*	3′-UTR	rs13223298	34,158,658	p14.3	G/T	0.08	−0.34 ± 0.08	8.7 × 10^−6^
15		*SEMA6D*	Intergenic	rs491274	46,577,328	q21.1	A/G	0.09	−0.31 ± 0.07	8.9 × 10^−6^
1	*RGL1*	*APOBEC4*	Intronic	rs6664058	183,687,143	q25.3	C/T	0.36	0.21 ± 0.05	1.2 × 10^−5^
7	*PKD1L1*	*TNS3*	Intronic	rs10276437	47,766,479	p12.3	C/T	0.78	0.21 ± 0.05	1.3 × 10^−5^
14		*VRK1*	Intergenic	rs4905587	97,213,901	q32.2	G/T	0.12	0.31 ± 0.07	1.4 × 10^−5^
1	*RGL1*	*APOBEC4*	Intronic	rs6424909	183,727,521	q25.3	A/G	0.36	0.21 ± 0.05	1.5 × 10^−5^
16		*HS3ST6*	5′-UTR	rs344363	1,922,547	p13.3	C/T	0.17	−0.26 ± 0.06	1.7 × 10^−5^
20	*PMEPA1*	*ZBP1*	Intronic	rs6025728	57,697,562	q13.31	C/T	0.63	0.20 ± 0.05	2.0 × 10^−5^
12	*CACNA1C*	*FKBP4*	Intronic	rs4765961	2,559,306	p13.33	C/T	0.82	0.23 ± 0.05	2.2 × 10^−5^

Chr, chromosome; A1/A2, baseline/effect allele; and EAF, effect allele frequency estimated from the reference panel 1000 G (European population). Beta and standard error (se) were estimated using the Gaussian linear model, taking into account the EGEA family structure and adjusted for principal components.

**Table 3 antioxidants-11-00802-t003:** A stratified analysis according to asthma status for the top three SNPs in the whole sample.

Chr	Gene	Nearest Gene	Genomic Location	Marker	A1/A2	EAF	Never-Asthmatics (n = 684)		Ever-Asthmatics (n = 532)		Homogeneity Test	
							**Beta ± se**	* **p** *	**Beta ± se**	* **p** *	**Khi^2^**	* **p** *
6	*BMP6*	*TXNDC5*	Intronic	rs270404	A/G	0.41	−0.20 ± 0.06	5.1 × 10^−4^	−0.20 ± 0.06	5.4 × 10^−4^	0.01	0.91
7		*BMPER*	3′-UTR	rs13223298	G/T	0.08	−0.42 ± 0.10	4.5 × 10^−5^	−0.23 ± 0.12	5.5 × 10^−2^	1.32	0.25
15		*SEMA6D*	Intergenic	rs491274	A/G	0.09	−0.36 ± 0.08	2.7 × 10^−5^	−0.23 ± 0.11	4.1 × 10^−2^	0.75	0.39

Chr, chromosome; A1/A2, baseline/effect allele; and EAF, effect allele frequency estimated from the reference panel 1000 G (European population). Beta and standard error (se) were estimated using Gaussian linear model, taking into account EGEA family structure and adjusted for principal components.

**Table 4 antioxidants-11-00802-t004:** The GWAS of FlOPs: the top 10 SNPs in never-smokers and in current smokers.

Chr	Gene	Nearest Gene	Genomic Location	Marker	Position bp (hg38)	Band	A1/A2	EAF	Beta ± se	*p*
Never-smokers (n = 604)										
6	*COL21A1*	*DST*	Intronic	rs17823624	56,348,021	p12.1	A/G	0.92	0.44 ± 0.08	2.3 × 10^−7^
15	*GABRG3*	*GABRA5*	Intronic	rs6606856	27,022,887	q12	C/T	0.70	−0.28 ± 0.05	4.1 × 10^−7^
5		*NUDT12*	Intergenic	rs2962642	104,131,010	q21.2	A/G	0.24	0.30 ± 0.06	4.4 × 10^−7^
10	*PAOX*	*MTG1*	Intronic	rs6537600	133,391,295	q26.3	C/T	0.88	0.40 ± 0.08	6.5 × 10^−7^
5		*NUDT12*	Intergenic	rs7725285	104,111,482	q21.2	G/T	0.75	−0.29 ± 0.06	1.5 × 10^−6^
10	*PAOX*	*MTG1*	Intronic	rs10776679	133,389,090	q26.3	C/T	0.05	−0.49 ± 0.10	1.6 × 10^−6^
8	*DEFB135*	*DEFB136*	Intronic	rs6985349	11,982,562	p23.1	C/T	0.06	−0.43 ± 0.09	2.0 × 10^−6^
8	*DEFB135*	*DEFB136*	Intronic	rs7004833	11,982,502	p23.1	A/G	0.06	−0.43 ± 0.09	2.0 × 10^−6^
3		*ZNF385D*	Intronic	rs1391857	20,857,452	p24.3	A/G	0.61	−0.27 ± 0.06	2.5 × 10^−6^
10	*ECHS1*	*PAOX*	Missense	rs1049951	133,370,622	q26.3	A/G	0.06	−0.41 ± 0.09	3.6 × 10^−6^
Current smokers (n = 275)										
6	*CRYBG1*	*ATG5*	Intronic	rs3851212	106,375,664	q21	A/G	0.94	−0.81 ± 0.13	2.4 × 10^−9^
12	*COL2A1*	*TMEM106C*	Intronic	rs1793958	47,998,650	q13.11	A/G	0.61	−0.40 ± 0.08	4.7 × 10^−7^
12		*PTPRO*	Intergenic	rs17174795	15,603,555	p12.3	G/T	0.87	−0.62 ± 0.12	9.2 × 10^−7^
15	*ISG20*	*AEN*	Intronic	rs8041687	88,655,329	q26.1	A/G	0.07	0.56 ± 0.11	1.1 × 10^−6^
3	*CMC1*	*AZI2*	Intronic	rs7641491	28,301,843	p24.1	A/G	0.53	0.39 ± 0.08	2.8 × 10^−6^
3	*CMC1*	*AZI2*	Intronic	rs13085075	28,309,712	p24.1	C/T	0.48	−0.39 ± 0.08	3.1 × 10^−6^
7		*CREB5*	Intergenic	rs10228137	28,848,469	p14.3	A/C	0.81	−0.43 ± 0.09	4.3 × 10^−6^
2		*LRP1B*	Intergenic	rs961109	142,396,452	q22.2	C/T	0.91	0.66 ± 0.14	4.6 × 10^−6^
4	*TENM3*	*DCTD*	Intronic	rs4557308	182,734,597	q35.1	A/G	0.26	0.41 ± 0.09	5.0 × 10^−6^
13		*ATP7B*	5′-UTR	rs4943040	51,919,816	q14.3	C/T	0.55	0.38 ± 0.08	5.2 × 10^−6^

Chr, chromosome; A1/A2, baseline/effect allele; and EAF, effect allele frequency estimated from the reference panel 1000 G (European population). Beta and standard error (se) were estimated using Gaussian linear model, taking into account EGEA family structure and adjusted for principal components.

**Table 5 antioxidants-11-00802-t005:** The stratified analysis according to asthma status for the top three SNPs, in never-smokers and in current smokers.

								Never-Asthmatics		Ever-Asthmatics		Homogeneity Test	
Chr	Gene	Nearest Gene	Genomic Location	Marker	Position bp (hg38)	A1/A2	EAF	Beta ± se	*p*	Beta ± se	*p*	khi²	*p*
	Never-smokers (n = 604)							n = 342		n = 262			
6	*COL21A1*	*DST*	Intronic	rs17823624	56 348 021	A/G	0.92	0.46 ± 0.11	2.9 × 10^−5^	0.43 ± 0.13	1.1 × 10^−3^	0.07	0.79
15	*GABRG3*	*GABRA5*	Intronic	rs6606856	27 022 887	C/T	0.70	−0.29 ± 0.08	2.4 × 10^−4^	−0.30 ± 0.08	1.7 × 10^−4^	0.00	0.97
5		*NUDT12*	Intergenic	rs2962642	104 131 010	A/G	0.24	0.36 ± 0.07	2.0 × 10^−6^	0.22 ± 0.09	1.5 × 10^−2^	1.89	0.17
	Current smokers (n = 275)							n = 140		n = 135			
6	*CRYBG1*	*ATG5*	Intronic	rs3851212	106 375 664	A/G	0.94	−0.79 ± 0.22	5.5 × 10^−4^	−0.85 ± 0.16	3.2 × 10^−7^	0.08	0.78
12	*COL2A1*	*TMEM106C*	Intronic	rs1793958	47 998 650	A/G	0.61	−0.38 ± 0.12	1.7 × 10^−3^	−0.42 ± 0.10	4.9 × 10^−5^	0.09	0.77
12		*PTPRO*	Intergenic	rs17174795	15 603 555	G/T	0.87	−0.85 ± 0.18	5.1 × 10^−6^	−0.36 ± 0.16	2.7 × 10^−2^	3.86	0.05

Chr, chromosome; A1/A2, baseline/effect allele; and EAF, effect allele frequency estimated from the reference panel 1000 G (European population). Beta and standard error (se) were estimated using Gaussian linear model, taking into account EGEA family structure and adjusted for principal components.

## Data Availability

Due to third-party restrictions, EGEA data are not publicly available. Please see the following URL for more information: https://egeanet.vjf.inserm.fr/index.php/en/contacts-en, accessed on 17 February 2022. Interested researchers should contact egea.cohorte@inserm.fr with further questions regarding data access.

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
