# Peer review of "Genome-Wide Association Study of Fluorescent Oxidation Products Accounting for Tobacco Smoking Status in Adults from the French EGEA Study"

_antioxidants, 2022, doi:10.3390/antiox11050802_

Round 1

Reviewer 1 Report

Authors

This paper presents a well-conducted but rather inconclusive GWAS analysis of plasma fluorescent oxidation products (FlOPs) in relation to asthma and to smoking.  

FlOPs are presented as a global biomarker measure for oxidative stress involved in CHD, CKD, and asthma.  You have conducted a straightforward GWAS with plasma samples from 1216 participants in the French EGEA cohort study, collected in the first follow-up (2003-2007); of the participants, 23% were current smokers and 44% were ever-asthma patients. GWAS was also analyzed separately for current smokers and never smokers.

From a total of 501,167 SNPs, the three top SNPs were evaluated as having particular relevance for asthma.  The laboratory and statistical methods are well described and appropriate. The mean FlOP levels were not hugely different:  93.8 RFU/ml for current smokers and 89.1 RFU/ml for never smokers. Higher levels were associated with smoking variables.  The top 10 SNPs associated with FlOP (Table 2; p<2.2E-05) were all below the statistical significance threshold of 1.3E-07 adjusted for multiple comparisons.

The top 3 SNPs associated with asthma status are the main focus of the paper: in BMP6, near BMPER, and between SQOR and SEMA6D on chromosomes 6, 7, and 15, respectively. Results with the two never-smoker and ever-smoker subsets did not confirm these SNPs from the larger study cohort and did not show impressive association with asthma.

Of potential interest:  Another study with long-banked serum samples is CARET, the beta-Carotene And Retinol Efficacy Trial for chemoprevention of lung cancers (and CHD) in men and women at high risk for lung cancers due to smoking and/or occupational asbestos exposure. The main results paper (NEJM 1996) showed not only no benefit, but also clear excess of lung cancers (1 additional lung cancer per 1000 participants per year, a startling increase above that in the placebo arm). The Fred Hutchinson Cancer Research Center has a publicly-available specimen bank. These chemoprevention clinical trial results triggered subsequent studies in ferrets that demonstrated that these vitamins/drugs caused increases in c-jun and c-fos cancer pathways.

Reviewer 2 Report

In general an interesting study in which genes that are associated with signals of oxidative stress are detected in a GWAS, with some additional evidence from eQTLs. I only have a slight concern on one step in the data analysis and would like to propose that some other steps in the analysis are specified in more detail.

major comments:
l. 132/133: Please explain: Why have these two different questions been asked, of which one is more precise than the other?

section 2.4: Why were all variables binned into categories or, in the case of bmi, even dichotomous variables? Wouldn't the continuous variables be more informative?

l. 171-176: I am skeptical about this procedure to avoid deviations from normality. I fear that this procedure could bias the results more than deviations from normal-distribution assumptions. Or is there a reference in which this approach has been shown to be preferable and statistically sound?

l. 185-188: Please give a clear specification about the statistical models and methods here, e.g. which types of GLMs and robust variance estimators and how was the PCA carried out.

l. 370/371: Can you give a (multiple-testing corrected) p-value for this association between SNP and methylation level?

minor comments:
l. 33: commas: "..this study, which is the first GWAS of FlOPs, brings.."
l. 61: Such approach -> Such an approach
l. 156: please clarify: does "first quartile (Q1) – third quartile (Q3)" refer to the interquartile range or to the two values? (In the results part it becomes clear that it is the latter.)
l. 173: "using Kolmogorov test" -> "using the Kolmogorov test" or "using Kolmogorov tests"
l. 191/192 splitted -> split

Reviewer 3 Report

The authors present a GWAS for FlOPs in a French cohort. The rationale for investigating FlOPs in this way is sound, but the low sample size (power?) and lack of independent validation of the findings are major points of concern. Is this phenotype available in any other population cohort, or even the UK Biobank? For either replication or to perform a meta-analysis - this would greatly improve the current study.

Minor comments:

  1. Please include sample sizes or numbers removed in the text. e.g. how many were excluded because CV was > 20% for the FlOP measure?
  2. Please provide details of the SNP QC performed, details of the imputation methods and number of individuals (were any removed due to poor SNP data?). At the very least a reference to published details, if applicable, is warranted. Ethnic checks?
  3. Given the phenotype was uniformly measured in the entire sample, why remove participants with |z-score| >2 ? How many individuals were removed in this process? I am concerned that actually this process is removing information and that the sample probably doesn't require so much processing. Please justify.
  4. The sub-samples are NOT independent of the full data set. At least, not independent as far as a genetic analysis is concerned. The sample size is small overall, and these sub-samples are even smaller, further reducing the statistical power to identify true variants. How were the two sub-samples grouped (line 270-273)? Doesn't this return us to the full sample? Details missing from methods.
  5. Most GWAS produce Manhattan plots, and genomic inflation estimates - please consider including/mentioning. In particular, lambda is a key value of interest, reflecting the quality of the results.
  6. Recommend the authors consider using a platform such as FUMA to gain more insight into the significant SNPs
  7. Given small sample sizes, and lack of independence in sub samples, I would suggest focusing on the overall analysis and move subsamples to the supplementary (or remove). 

Other:

  • Table 1 footnote contains a header at the end?
  • Section 3.2 intro is repeated from the Methods section.
  • How are Previous and Next gene defined? Consider giving genomic location (intronic/exonic/intergenic) and nearest gene instead.

Round 2

Reviewer 2 Report

Many thanks to authors for appropriately addressing all my previous comments.

Author Response

We thank the reviewer for his kind comments and for allowing us to improve the manuscript with his very pertinent suggestions and comments.

Reviewer 3 Report

The authors have addressed most of my concerns with this revision. My major concern still remains - that this study has not been independently validated. I agree that the two subsamples are independent of each other. From a statistical genetic standpoint, it is customary to analyse genetic associations in different cohorts (independent samples), and to present the results of a meta-analysis (combining all cohorts together) or mega-analysis (if individual level data can be shared). The results from the meta/mega analysis must still be independently validated using different cohorts to those used in the original analysis, or else must state that the findings still need to be validated/replicated. Here, the authors have presented a mega-analysis and then used the two subsample (~cohort) analyses as independent validation. This is not correct or acceptable. The authors should present the overall results and state independent validation is still required. The two subsamples can not be used as independent validation of the mega analysis since that analysis consisted of all samples in the two subsamples.

Additionally, (minor), please state the reference panel that was used for imputation.
